

# Enhanced representation of soil NO emissions in the Community Multi-scale Air Quality (CMAQ) model

**Quazi Z. Rasool[1], Rui Zhang[1], Benjamin Lash[1*], Daniel S. Cohan[1], Ellen Cooter[2], Jesse Bash[2] and Lok N. Lamsal[3,4]**

[1]{Department of Civil and Environmental Engineering, Rice University, Houston, Texas, USA}

[2]{Computational Exposure Division, National Exposure Research Laboratory, Office of Research and Development, US Environmental Protection Agency, RTP, NC, USA}

[3]{Goddard Earth Sciences Technology and Research, Universities Space Research Association, Columbia, MD 21046, USA }

[4]{NASA Goddard Space Flight Center, Greenbelt, MD 20771, USA}

[*]{now at: School of Natural Sciences, University of California, Merced, CA}

*Correspondence to*:  Daniel Cohan (cohan@rice.edu)



## Abstract

Modeling of soil nitric oxide (NO) emissions is highly uncertain and may misrepresent its spatial and temporal distribution. This study builds upon a recently introduced parameterization to improve the timing and spatial distribution of soil NO emission estimates in the Community Multi-scale Air Quality (CMAQ) model. The parameterization considers soil parameters, meteorology, land use, and mineral nitrogen (N) availability to estimate NO emissions. We incorporate daily year-specific fertilizer data from the Environmental Policy Integrated Climate (EPIC) agricultural model to replace the annual generic data of the initial parameterization, and use a 12 km resolution soil biome map over the continental US. CMAQ modeling for July 2011 shows slight differences in model performance in simulating fine particulate matter and ozone from IMPROVE and CASTNET sites and $NO_2$ columns from Ozone Monitoring Instrument (OMI) satellite retrievals. We also simulate how the change in soil NO emissions scheme affects the expected $O_3$ response to projected emissions reductions.



## 1 Introduction

Nitrogen oxides ($NO_x$=NO+$NO_2$) play a crucial role in tropospheric chemistry. Availability of $NO_x$ influences the oxidizing capacity of the troposphere as $NO_x$ directly reacts with hydroxyl radicals (OH) and catalyzes tropospheric ozone ($O_3$) production and destruction (Seinfeld and Pandis, 2012). $NO_x$ also affects the lifetime of reactive greenhouse gases like $CH_4$ by influencing its dominant oxidant OH (Steinkamp and Lawrence, 2011), thus affecting the Earth's radiative balance (IPCC, 2007). $NO_x$ also influences rates of formation of inorganic particulate matter (PM) (Wang et al., 2013) and organic PM (Seinfeld and Pandis, 2012).

Soil $NO_x$ emissions accounts for ~15-40 % of the tropospheric $NO_2$ column over the continental United States (CONUS), and up to 80% in highly N fertilized rural areas like the Sahel of Africa (Hudman et al., 2012). The estimated amount of nitric oxide (NO) emitted from soils is highly uncertain, ranging from 4-15 Tg-N $yr^{-1}$, with different estimates of total global $NO_x$ budget also showing a mean difference of 60-70% (Potter et al., 1996; Davidson and Kingerlee, 1997; Yienger and Levy, 1995; Jaeglé et al., 2005; Stavrakou et al., 2008; Steinkamp and Lawrence, 2011; Miyazaki et al., 2012; Stavrakou et al., 2013; Vinken et al., 2014). Soil $NO_x$ is mainly emitted as NO through both microbial activity (biotic/enzymatic) and chemical (abiotic/non-enzymatic) pathways, with emission rates varying as a function of meteorological conditions, physicochemical soil properties, and nitrogen (N) inputs from deposition and fertilizer or manure application (Pilegaard, 2013; Hudman et al, 2012). The fraction of soil N emitted as NO varies with meteorological and soil conditions such as temperature, soil moisture content, and pH (Ludwig et al., 2001; Parton et al., 2001; van Dijk et al., 2002; Stehfest and Bouwman, 2006).

Different biome types, comprised of vegetation and soil assemblages exhibit different NO emission factors under different soil conditions and climate zones. One of the early attempts to stratify soil NO based on different biomes by Davidson and Kingerlee (1997) involved compiling over 60 articles and 100 field estimates. They clearly identified biomes associated with low NO emissions like swamps, tundra, and temperate forests, and those with high soil NO fluxes like tropical savanna/woodland and cultivated agriculture. For instance, high soil NO fluxes were observed in croplands, savannahs or woodlands, N-rich temperate forests and even boreal/tropical forests with low $NO_2^-$ availability in warm conditions and acidic soil (Kesik et al., 2006; Cheng et al., 2007; Su et al., 2011). This approach, however, fails to capture within-



biome variation in NO emissions (Miyazaki et al., 2012; Vinken et al., 2014). Steinkamp and
Lawrence (2011) more recently compiled worldwide emission factors from a dataset consisting
of 112 articles with 583 field measurements of soil $NO_x$ covering the period from 1976 to 2010,
and regrouped them into 24 soil biome type based on MODIS land cover category as well as
Köppen climate zone classifications (Kottek et al., 2006).
N deposition can be a significant driver of soil NO emissions in N-limited settings or near strong
N emissions sources, where both wet and dry deposition of N species act like an additional
fertilizer source (Yienger and Levy, 1995; Hudman et al., 2012). The response of soil $NO_x$
emission to N deposition varies as a function of soil N status and land-use history of the land
use/biome type. Mature forests for instance with already high initial soil N due to higher
mineralization rates give higher soil NO flux than rehabilitated and disturbed ones (Zhang et al.,
2008). In agricultural soils, N deposition is a leading contributor to the inorganic N pool that
eventually contributes to soil NO emissions (Liu et al., 2006; Pilegaard, 2013).
Fertilizer (organic and inorganic) application represent controllable influences on soil N
emissions (Pilegaard, 2013) and are leading sources of reactive nitrogen (N) worldwide
(Galloway and Cowling, 2002). U.S. fertilizer use increased by nearly a factor of 4 from 1961 to
1999 (IFIA, 2001). Soil NO emissions increase with rising fertilizer application, with conversion
rate of applied fertilizer N to $NO_x$ being up to ~ 11% (Williams et al., 1988; Shepherd et al.,
1991). Open and closed chamber studies have shown increasing fertilizer application to increase
both NO and $N_2O$ fluxes simultaneously, but with variability in $NO/N_2O$ emission ratio
(Harrison et al., 1995; Conrad, 1996; Veldkamp and Keller, 1997).
Meteorological conditions influence soil NO emission rates. Large pulses of biogenic NO
emissions often follow the onset of rain after a dry period (Davidson, 1992; Scholes et al., 1997;
Jaeglé et al., 2004; Hudman et al., 2010). Soil NO pulsing events occur when water stressed
nitrifying bacteria, which remain dormant during dry periods, are activated by the first rains and
start metabolizing accumulated N in the soil. NO pulses of up to 10–100 times background levels
typically last for about 1–2 days (Yienger and Levy, 1995; Hudman et al., 2012).
Adsorption onto plant canopy surfaces can reduce the amount of soil NO emissions entering the
broader atmosphere. Yienger and Levy (1995) (YL) soil NO scheme followed a Canopy
Reduction Factor (CRF) approach (Wang et al., 1998) to account for the reduction of soil NO



emission flux *via* stomatal or cuticle exchange as a function of dry deposition within the canopy
on a global scale.
Contemporary air quality models such as the Community Multi-scale Air Quality (CMAQ)
model most often use an adaptation of the YL scheme to quantify soil NO emissions as a
function of fertilizer application, soil moisture, precipitation and CRF (Byun and Schere, 2006).
However, YL has been found to underestimate emissions rates inferred from satellite and ground
measurements by a factor ranging from 1.5 to 4.5, and to misrepresent some key spatial and
temporal features of emissions (Jaeglé et al., 2005; Wang et al., 2007; Boersma et al., 2008; Zhao
and Wang, 2009; Lin, 2012; Hudman et al., 2012; Vinken et al., 2014). This overall
underestimation can be attributed to several uncertainties in the modeling settings, such as
inaccurate emissions coefficients, poor soil moisture data, deriving soil temperatures from
ground air temperatures, neglecting nitrogen deposition and outdated fertilizer application rates
(Yienger and Levy, 1995; Jaeglé et al., 2005; Delon et al., 2007; Wang et al., 2007; Boersma et
al., 2008; Delon et al., 2008; Hudman et al., 2010; Steinkamp and Lawrence, 2011; Hudman et
al., 2012).
The Berkley Dalhousie Soil NO Parameterization (BDSNP) scheme, originally implemented by
Hudman et al. (2012) in the GEOS-Chem global chemical transport model, outperforms YL by
better representing biome type, the timing of emissions, and actual soil temperature and moisture
(Hudman et al., 2010).

Our approach builds upon BDSNP by using the Environmental Policy Integrated Climate (EPIC)
biogeochemical model for dynamic representation of the soil N pool on a day-to-day basis. EPIC
is a field scale biogeochemical process model developed by the United States Department of
Agriculture (USDA) to represent plant growth, soil hydrology, and soil heat budgets for multiple
soil layers of variable thickness, multiple vegetative systems and crop management practices
(Cooter et al., 2012). EPIC can model up to 1 sq. km (100 ha) spatially and on a daily time scale
(CMAS, 2015). EPIC simulations are compatible with spatial and temporal scale of CMAQ as
well (Bash et al., 2013). EPIC accounts for different agricultural management scenarios, accurate
simulation of soil conditions and plant growth to produce plan demand-driven fertilizer estimates
for BDSNP (Cooter et al., 2012; Bash et al., 2013).



Baseline soil NO emission rate for each location (Hudman et al., 2012; Vinken et al., 2014), use
a new soil biome map with finer-scale representation of land cover systems consistent with
typical resolution of a regional model. We also built an offline version of BDSNP (stand-alone
BDSNP), which can use benchmarked inputs from the CMAQ and allows quick diagnostic based
on soil NO estimates for sensitivity analysis (Supplementary material Section S.2).


## 124  2   Methodology


### 126  2.1 Implementation of advanced soil NO parameterization in CMAQ

### 127  2.1.1 Land surface model (LSM)

Our implementation of the BDSNP soil NO parameterization in CMAQ uses Pleim-Xiu Land
Surface Model (Pleim and Xiu, 2003). Compared to the coarser LSM in GEOS-Chem (Bey et al.,
2001), Pleim-Xiu provides finer-scale estimates of soil moisture and soil temperature based on
solar radiation, temperature, Leaf Area Index (LAI), vegetation coverage, and aerodynamic
resistance. The rich amount of information available from the Pleim-Xiu LSM enables refined
representation of soil moisture and soil temperature for implementation in soil NO
parameterization.

### 135  2.1.2 Canopy reduction factor

The original implementation of BDSNP in GEOS-Chem did not provide specific spatial-
temporal variation of CRF in each modeling grid, but used a monthly average CRF from Wang
et al. (1998). Wang et al. (1998) included an updated CRF as part of their implementation of YL
into GEOS-Chem. This CRF is based on wind speed, turbulence, canopy structure, deposition
constants, and other physical variables. In the GEOS-Chem implementation of BDSNP, this CRF
reduced the flux by ~ 16%, from 10.7 Tg-N yr$^{-1}$ above soil to 9 Tg-N yr$^{-1}$ above canopy
(Hudman et al., 2012).



Our BDSNP implementation for CMAQ uses the same approach of integrating CRF as used in
Wang et al. (1998) with the biome categorization based on Steinkamp and Lawrence (2011) and
Köppen climate classes (Kottek et al. 2006) in the soil $NO_x$ parametrization itself.

### 2.1.3 Fertilizer

YL in CMAQ assumed a linear correlation between fertilizer application and its induced
emissions over general growing season, May-August in the Northern Hemisphere and
November-February in the Southern Hemisphere (Yienger and Levy, 1995) rather than peaking
near the time of fertilization at the beginning of the local growing season. This likely caused
inaccurate temporal representation of fertilizer driven emissions in certain regions (Hudman et
al., 2012). The GEOS-Chem implementation of BDSNP applied a long-term average fertilizer
application with a decay term after fertilizer is applied. Constant fertilizer emissions neglect an
important phenomenon: applying fertilizer during a dry period when neither plants nor bacteria
may have the water available to use it may result in a large pulse when the soil is eventually re-
wetted (Pilegaard, 2013). Such dry spring N fertilizer application can be quite significant,
especially in the mid-west and southern plains in the US (Cooter et al., 2012). The current
fertilizer data used for the BDSNP is scaled to global 2006 emissions by Hudman et al. (2012)
using a spatial distribution for year 2000 from Potter et al. (2010). This global database reported
by Potter et al. (2010) is already 8 years out of date in magnitude and 14 years out of date for
relative distribution, and has relatively coarse resolution based on out-of-date long term average
(national-level fertilizer data from 1994 to 2001). Using recent fertilizer application information
is essential to soil NO estimates given the fact that N fertilizer is the major contributor to plant
nutrient use in US, and its share has been increasing from 11,535 thousand short tons in 2001 to
12,840,000 short tons in 2013 (USDA ERS, 2013). Our implementation of BDSNP into CMAQ
is designed to enable updates by subsequent developers to use new year- and location- specific
fertilizer data. We use the Fertilizer Emission Scenario Tool for CMAQ (FEST-C v1.1,
http://www.cmascenter.org) to incorporate EPIC fertilizer application data into our CMAQ runs.

### 2.1.4 N Deposition



YL in CMAQ neglects nitrogen deposition, which can result in an 0.5 Tg/yr underestimation in
soil $NO_x$ globally (~5%) (Hudman et al., 2012). The implementation of the EPIC model in
FEST-C inputs oxidized and reduced form of N deposition directly into soil ammonia and nitrate
pools each day.  In Our implementation of BDSNP, these daily time series derive from previous
CMAQ simulation.  Inclusion of this deposition N source acts to reduce the simulated plant-
based demand for additional N applications.
**2.1.5 Formulation of soil NO scheme**
Figure 1 provides the flow chart of the BDSNP scheme implementation, which has the option to
run in-line with CMAQ, or offline as a stand-alone emissions parameterization. Static input files
in Hudman et al. 2012 BDSNP implementation (labelled as 'old' in Fig. 1) such as those giving
soil biome type with climate zone and global fertilizer pool are needed to determine the soil base
emission value at each modeling grid. The Meteorology-Chemistry Interface Processor (MCIP)
(Otte and Pleim, 2010) takes outputs from a meteorological model such as Weather Research and
Forecasting (WRF) model (Skamarock et al., 2008) to provide a complete set of meteorological
data needed for emissions and air quality simulations.
There are seven key input environment variables and two key output environment variables in
our implementation of BDSNP. Table S1 lists their names and corresponding functionalities.
Our implementation of the BDSNP soil NOx emission, $S_{NO_x}$ in CMAQ multiplies a base
emission factor ($A$) by scaling factors dependent on soil temperature ($T$) and soil moisture ($\theta$),
i.e., $f(T)$, $g(\theta)$ and a pulsing term ($P$) (equation 1). The base emission factor depends on biome
type under wet or dry soil conditions. The pulsing term depends on the length of the dry period,
rather than the accumulated rainfall amount considered by YL. The CRF term estimate the
fractional reduction in soil $NO_x$ flux due to canopy resistance.



$S_{NO_x}\ Flux(\frac{ng\,N}{m^2 s}) =$
$A'_{biome}(N_{avail}) \times f(T) \times g(\theta) \times P(l_{dry}) \times CRF(LAI, Meterology, Biome)$     (1)
$A'_{biome} = A_{biome} + N_{avail} \times \bar{E}$     (2)
$N_{avail}(t) = N_{avail\ Fert}(0) \times e^{-\frac{t}{\tau_1}} + F \times \tau_1 \times \left(1 - e^{-\frac{t}{\tau_1}}\right) + N_{avail\ Dep}(0) \times e^{-\frac{t}{\tau_2}} + D \times \tau_2 \times$
$(1 - e^{-\frac{t}{\tau_2}})$

198   (3)

Fertilizer and deposition both contribute to modifying the $A'_{biome}$ emissions coefficients for each
biome. Available nitrogen ($N_{avail}$) at time $t$ from fertilizer and deposition is multiplied by
emission rate, $\bar{E}$, based on the observed global estimates of fertilizer emissions ($\sim 1.8$ Tg-N yr[-1])
by Stehfest and Bouwman (2006) and added to biome specific soil NO emission factors ($A_{biome}$)
from Steinkamp and Lawrence (2011) to give the net base emission factor ($A'_{biome}$) (Eq. (2) and
Eq. (3)). The resulting $A'$ is multiplied by the meteorological scaling or response factors: $f(T)$,
$g(\theta)$, and $P(l_{dry})$ as in Eq. (1). The soil temperature response or scaling factor $f(T)$ is simplified to
be exponential everywhere. NO flux now depends on soil moisture ($\theta$) instead of rainfall, and it
increases smoothly to a maximum value before decreasing as the ground becomes water
saturated. In Eq. (3), $F$ is fertilization rate (kg ha[-1]), $D$ is the wet and dry deposition rate (kg ha[-1])
considered as an additional fertilization rate, and $\tau$ is decay time, which is 4 months for fertilizer
($\tau_1$) and 6 months for deposition ($\tau_2$) (Hudman et al. 2012).
BDSNP uses a Poisson function to represent the dependence of emission rates on soil moisture
($\theta$), where the parameters '$a$' and '$b$' vary for different climates such that the maximum of the
function occurs at $\theta = 0.2$ for arid soils and $\theta = 0.3$ otherwise (Hudman et al. 2012). We adopt
the same approach in CMAQ, as follows:
$f(T) * g(\theta) = e^{0.103*T} * a * \theta * e^{-b*\theta^2}$     (4)
The pulsing term depends on the length of the dry period ($l_{dry}$) and a change in soil moisture
instead of on the amount of precipitation (Hudman et al., 2012).



The pulsing term for emissions when rain follows a dry period is
$$P(l_{dry}, t) = \left[13.01 * \ln(l_{dry}) - 53.6\right] * e^{-c*t} \tag{5}$$
In this equation, $l_{dry}$ is the length of the dry period that preceded the rain and $c = 0.068$ hour$^{-1}$
defines the exponential decay of the pulse.
Beyond this basic implementation of the above stated BDSNP framework into CMAQ, there
were major modifications (highlighted as 'new' in Fig. 1) in the form of: a) updating biome map
consistent with CMAQ, b) incorporating year- and location- specific fertilizer data using EPIC
outputs and c) development of a standalone BDSNP module. Our work focuses on those
developments discussed in detail in the sections to follow.

## 2.2 Soil biome map over CONUS

The original implementation of BDSNP used the global soil biome data from the GEOS-Chem,
with emission factors for each biome under dry/wet conditions taken from Steinkamp and
Lawrence (2011) (Appendix Table A1). Our implementation in CMAQ uses a finer resolution
(12 km) soil biome map over CONUS. The map is generated from the 30-arc-second
(approximately 1 kilometer) NLCD40 (National Land Cover Dataset) for 2006, with 40 land
cover/land use classifications. A mapping algorithm table (see Appendix Table A2) was created
to connect the land use category to soil biome type (Table A1) based on best available
knowledge. For the categories with identical names, such as 'evergreen needleleaf forest',
'deciduous needleleaf forest', 'mixed forest', 'savannas' and 'grassland', the mapping is direct.
Categories in NLCD40, which are subsets of the corresponding biome category, are consolidated
into one category by addition. For example, 'permanent snow and ice' and 'perennial ice-snow'
in NLCD40 are combined to form 'snow and ice'; 'developed open space', 'developed low
intensity', 'developed medium intensity', and 'developed high intensity' are added to form
'urban and built-up lands'. For the categories appearing only in NLCD40, the mapping algorithm
is determined by referring to the CMAQ mapping scheme, available in Cross-Section and
Quantum Yield (CSQY) data files in the CMAQ coding. One such case is to map 'lichens' and



'moss' in NLCD40 to the category 'grassland' in soil biome. Furthermore, a model resolution
compatible Köppen climate zone classification (Kottek et al., 2006) was added to allocate
different emission factor for the same biome type e.g. to account for different altitudes of
'grassland' at different locations. There are five climate zone classifications, namely A:
equatorial, B: arid, C: warm temperature, D: snow, E: polar. A 12 km CONUS model resolution
climate zone classification map (see Figure 2) was created using the Spatial Allocator based on
the county level climate zone definition as the surrogate based on a dominant land use,
(http://koeppen-geiger.vu-wien.ac.at/data/KoeppenGeiger.UScounty.txt).
Figure 2 compares the 24 soil biome map with 0.25 degree resolution from the GEOS-Chem
settings to the new 12 km resolution soil biome map we created here for CMAQ. Table A2 gives
the biome type names with corresponding climate zones.
The classification of simulation domain into arid and non-arid region with consistent resolution
is also included in our implementation. Figure B1 shows the distribution of arid (red) and non-
arid (blue) regions. For the modeling grid classified as 'arid' region, the maximum moisture
scaling factor corresponds to the water-filled pore space ($\theta$) value equal to 0.2; while for the
'non-arid' modeling grid, the   maximum moisture scaling factor corresponds with $\theta$=0.3
(Hudman et al., 2012).

### 262   2.3 Representation of fertilizer N

We implemented two approaches for representing fertilizer N. The first approach regrids
fertilizer data from the global GEOS-Chem BDSNP implementation (Hudman et al. 2012) to our
12 km resolution CONUS domain. That scheme uses the global fertilizer database from Potter et
al. (2010) and assumed 37% of fertilizer and manure N is available (1.8 Tg-N yr$^{-1}$) for potential
emission. Figure B2 provides the day-by-day variation of total N remaining due to fertilizer
application over CONUS during a year, and shows the typical cycle between growing season and
non-growing season. The Potter data, however, are a decade old and at coarse resolution for
county-level in US.
Our second approach (Figure 3) uses the EPIC model as implemented in the FEST-C tool
(Cooter et al. 2012) to provide a dynamic representation of fertilizer applications for a specific
growing season. FEST-C (v1.1) generates model-ready fertilizer input files for CMAQ.  . Use of





FEST-C/EPIC instead of soil emissions from YL scheme has been shown to improve CMAQ
performance for nitrate and ammonia in CONUS (Bash et al., 2013). The BELD4 tool in FEST-
C system was used to provide the crop usage fraction over our domain. We summed FEST-C
data for ammonia, nitrate and organic, T1_ANH3, T1_ANO3 and T1_AON respectively in kg-
N/ha, to give a total soil N pool for each of 42 simulated crops (CMAS, 2015). This daily crop-
wise total soil N pool was then weighted by the fraction of each crop type at each modeling grid
to get a final weighted sum total soil N pool usable in BDSNP. CMAQ v.5.0.2 can be run with
in-line biogenic emissions, calculated in tandem with the rest of the model. Since the EPIC N
pools already include N deposition, we designed our soil NO emissions module to be flexible in
recognizing whether it is using fertilizer data such as Potter et al. (2010) that does not include
deposition or EPIC that does.
Figure 4 compares the FEST-C derived N fertilizer map and the default coarser resolution long-
term average fertilizer map from Potter. While the spatial patterns are similar, EPIC provides
finer resolution and more up-to-date information.

## 2.4 Model configurations and data use for model evaluations

The CMAQ domain settings for CONUS as provided by the EPA were used to simulate the
whole month of July in 2011. July corresponds to the month of peak flux for soil nitrogen
emissions in the United States (Williams et al., 1992; Cooter et al., 2012; Bash et al., 2013) and
is an active period for ozone photochemistry (Cooper et al., 2014; Strode et al., 2015).
A ten day (21 June-30 June, 2011) spin-up time was used to minimize the influence from initial
conditions. The domain consisted of 396 columns, 246 rows, 26 vertical layers, and 12 km
rectangular cells using a Lambert Conformal Projection over North America. This configuration
was consistent throughout the WRF-BDSNP-CMAQ modeling framework (see Figure 1).
Meteorology data were produced through the WRF Model nudged to National Centers for
Environmental Prediction (NCEP) and National Center for Atmospheric Research Reanalysis
(NARR) data, which is comprised of historical observations and processed to control quality and
consistency across years by National Oceanic and Atmospheric Administration (NOAA).



Emissions were generated using the Sparse Matrix Operator Kernel Emissions (SMOKE) model
(CMAS, 2014) and 2011NEIv1.

We applied CMAQ with three sets of soil NO emissions: a) Standard YL soil NO scheme, b)
BDSNP scheme with Potter et al. (2010) fertilizer data set and biome mappings from GEOS-
Chem, and c) BDSNP scheme with EPIC 2011 data and new biome mappings. Within these
three cases, we simulated the impact of anthropogenic $NO_x$ reductions applied to all contributing
source sectors listed in the 2011 National Emission Inventory (NEI). For this purpose, we
considered the baseline $NO_x$ reduction scenario from 2011 to 2025 that EPA's Regulatory
Impact Analysis (RIA) determined for Business as Usual (BAU) in the CONUS domain (Figure
2A-1, Table 2A-1 in https://www3.epa.gov/ttn/ecas/docs/20151001ria.pdf). Table 1 gives a full
list of modeling configurations settings used for achieving the above-mentioned simulations.
Model simulations were evaluated against the following in situ and satellite-based data: 16
USEPA Clean Air Status and Trends Network (CASTNET) sites for MDA8 $O_3$
(www.epa.gov/castnet), 9 Interagency Monitoring of Protected Visual Environments
(IMPROVE) sites for daily average $PM_{2.5}$ (Malm et al., 1994), and NASA's OMI retrieval
product for tropospheric $NO_2$ column (Bucsela et al., 2013; Lamsal et al., 2014). Fig. 5 shows the
spatial distribution of the ground sites used for validation of modeled estimates. The selected
ground sites for model validation are mostly based in agricultural regions with intense fertilizer
application rate and high NO fluxes, specifically the Midwest, southern plains, and San Joaquin
Valley.

We also simulated three sensitivity cases for the same time period and domain with the offline
soil NO module: a) NLCD40 based (new) biome vs GEOS-Chem based (old) biome (using EF1
in Table A1), b) EPIC 2011 vs Potter data and, c) Global mean biome emission factor (EF1 in
Table A1) vs North American mean emission factor (EF3 in Table A1) (Supplementary material
Section S.3).



## 3  Results and Discussion

### 3.1  Spatial distribution of nitrogen fertilizer application and soil NO emissions over CONUS

We demarcated the CONUS domain into six sub-domains (Figure 6) to analyze model outputs. The updated BDSNP model and EPIC fertilizer result in higher soil NO emission rates than YL and Potter. Emissions increase by a factor ranging from 1.8 to 2.8 in shifting from YL to BDSNP, even while retaining the Potter fertilizer data and original biome map, indicating that the shift from YL to BDSNP scheme is the largest driver of the increase in emissions estimates. EPIC and the new biome dataset further increase emissions over most of CONUS, except for the southwest region. In Midwest and Western US, the new biome map identified more cropland and shifted some grasslands to other land cover types such as forests, savannah and croplands, which exhibit higher soil NO emissions (Figure 2; Table A1). The Midwest region is characterized with the highest emission rate due to its abundant agricultural lands with high fertilizer application rates (Figure 4).

### 3.2 Evaluation of CMAQ $NO_2$ with satellite OMI $NO_2$ observations

The standard (version 2.1) OMI tropospheric $NO_2$ column observations from NASA's Aura satellite as discussed in Bucsela et al. (2013) and Lamsal et al. (2014) were used for comparison with our modelled $NO_2$ vertical columns. To enable comparison, the quality-assured, clear-sky (cloud radiance fraction < 0.5) OMI $NO_2$ data were gridded and projected to our domain by using ArcGIS 10.3. CMAQ modelled $NO_2$ column densities in molecules per $cm^2$ were derived using vertical integration and extracted for 13:00-14:00 local time, corresponding to the time of OMI measurements.

We compared CMAQ simulated tropospheric $NO_2$ columns with OMI product for regions showing highest sensitivity in soil NO switching from YL to BDSNP: Midwest, San Joaquin Valley in California and central Texas (see Appendix Figure B3). Switching from YL to our updated BDSNP ('new') module improved agreement with OMI $NO_2$ columns in central Texas





but over-predicts column $NO_2$ in the San Joaquin Valley and Midwest (Figure 7). Even the YL
estimate was higher than OMI by a factor of two in the Midwest (Figure 7).

## 3.3 Evaluation with $PM_{2.5}$ and ozone observations

Model results are compared with observational data from IMPROVE monitors for $PM_{2.5}$ and
CASTNET monitors for ozone. We first compute differences between ozone and $PM_{2.5}$ estimates
from the three simulation cases to identify sites influenced by the choice of soil NO scheme
during our July 2011 episode (Figures 8 and 9). These highlights nine IMPROVE sites for $PM_{2.5}$
and 16 CASTNET sites for ozone (Figures 5, 8 and 9) where CMAQ results are sensitive to soil
NO changes (Figure 6).
Statistical comparisons of modeled and observed daily average $PM_{2.5}$ at the nine IMPROVE sites
are provided in Table 2. Mean Absolute Gross Error (MAGE) and Root Mean Square Error
(RMSE) improved from 2.8 to 2.7 ug/m$^3$ and 3.4 to 3.3 ug/m$^3$ respectively when moving from
YL to BDSNP with the new inputs. Both Pearson's and Spearman's ranked correlation
coefficient (R) shows no significant change when soil NO module in CMAQ is switched from
YL to BDSNP (Potter with old biome) and BDSNP (EPIC with new biome) (Tables 2). Use of
the ranked correlation coefficient minimizes the impact of spurious correlations due to outliers
but does not affect the analysis.  Switching from YL to our updated BDSNP ('new') module
shows that the predicted versus observed fit becomes slightly closer to 1:1 (Figure 10).
Numerical Mean Bias (NMB) and Numerical Mean Error (NME) improve from -28.5% to -
26.4% and 34.6% to 33.6%, respectively.
In contrast to the $PM_{2.5}$ results, the updated soil NO scheme yields mixed impacts on model
performance for maximum daily average 8-hour (MDA8) ozone at the targeted 16 CASTNET
sites (Table 3 and Figure 11). For the 11 agricultural/prairie sites, replacement of YL with
BDSNP with new inputs increases NMB from 7.6% to 14.1% and NME from 15.7 to 19.3%
(Table 3). The excess ozone may occur because FEST-C does not account for the loss of
fertilizer N to the water stream ("tile drainage") in wet conditions (Dinnes et al., 2002). Hudman
et al. (2012) suggested $\theta = 0.175$ (m$^3$/m$^3$) as threshold below which dry condition occur. During
July 2011, in Midwest monthly mean soil moisture ($\theta_{mean}$, m$^3$/m$^3$) is mostly > 0.175, indicating



possibility of wet conditions (Fig. S5). This can also be due to known wet bias in WRF simulated
meteorology i.e. more perceived precipitation than observed (Zhang et al., 2009) which may
result in high NO emissions in moist soils. Overestimation of $O_3$ is due to higher NO emissions,
as these regions comprise of mostly $NO_x$ limited rural locations.
At the California CASTNET sites, BDSNP enhances model performance in simulating observed
MDA8 ozone (Table 3). This can be seen in the NMB, NME, MAGE, and RMSE comparisons
between YL and BDSNP, though updating BDSNP to the newer inputs does not enhance
performance (Table 3).

## 3.4 Impact of soil NO scheme on ozone sensitivity to anthropogenic $NO_x$ perturbations



We analyzed how the choice of soil NO parameterization affects the responsiveness of ozone to
reductions in anthropogenic $NO_x$ emissions. We applied emission perturbation factors based on
the 5.7 million ton reduction in baseline anthropogenic $NO_x$ emissions from 2011 to 2025 that
US EPA simulated in its latest RIA (U.S. EPA, 2015). Table 4 gives the perturbation factors we
used to obtain baseline anthropogenic $NO_x$ emissions for 2025 over all contributing sectors as
listed from NEI 2011. Since our simulation is for July 2011 over CONUS, we used these
perturbation factors rather than the net reductions in RIA to scale emissions in a similar pattern
as given in RIA for annual baseline perturbations from 2011 to 2025 with BAU.

Shifting from YL to the BDSNP soil NO scheme reduces the sensitivity of MDA8 $O_3$ to
anthropogenic $NO_x$ perturbations. The impacts are greatest in California and the Midwest, where
shifting to BDSNP can reduce the expected impact of the anthropogenic $NO_x$ reductions by ~ 1
to 1.5 ppbV. Changing the inputs within the BDSNP scheme has a smaller impact (Figure 12).
Our results imply that the higher soil NO emissions from our updated BDSNP module shifts the
ozone photochemistry to a less strongly $NO_x$-limited regime.



## 4 Conclusions

Our BDSNP implementation represents a substantial update from the YL scheme for estimating soil NO in CMAQ. Compared to the previous implementation of BDSNP in global GEOS-Chem model, our implementation in CMAQ incorporated finer-scale representation of its dependence on land use, soil conditions, and N availability. This finer resolution and updated biome and fertilizer data set resulted in higher sensitivity of soil NO to biome emission factors. Our updated BDSNP scheme (EPIC and new biome) predicts slightly higher soil NO than the inputs used in GEOS-Chem, primarily due to the use of 2011 daily EPIC/FEST-C fertilizer data and fine resolution NLCD40 biomes (Figure 6).

Sensitivities to different input datasets were examined using our standalone BDSNP module to reduce computational cost. Switching from GEOS-Chem biome to new NLCD40 biome drops soil NO in the northwest and southwest portions of our domain due to the finer resolution biome map exhibiting lower emission factors in those regions. Replacing fertilizer data from Potter et al. (2010) with an EPIC 2011 dataset increased soil NO mostly in the Midwest (Supplementary material Figure S4).

We compared tropospheric $NO_2$ column densities output from our CMAQ runs with the three inline soil NO schemes to OMI observations as spatial average over regions sensitive to switch from YL to our updated BDSNP scheme. Temporal average of OMI and CMAQ simulated $NO_2$ column densities was done over the OMI overpass time (13:00-14:00 local time) for July 2011 monthly mean. Figure 7 summarizes tropospheric $NO_2$ column density comparisons between model and OMI satellite observation for aforementioned sensitive regions. Central Texas showed improvement with switch from YL to our BDSNP ('new') scheme. For July 2011, central Texas and San Joaquin Valley exhibit relatively dry soil conditions, whereas the Midwest was mostly wet (Supplementary material Figure S5). Even with similar conditions as central Texas, San Joaquin region shows overall degradation. Overestimation of simulated $NO_2$ columns up to twice of OMI over Midwestern US and San Joaquin valley for summer episodes has been exhibited earlier as well (Lamsal et al., 2014). Several factors, such as spatial inhomogeneity within OMI pixels and possible errors arising from the stratosphere-troposphere separation scheme and air mass factor calculations, can be attributed to this overestimation. Retrieval difficulties in complex terrain may explain the discrepancies in $NO_2$ column over San Joaquin Valley even



though it shows slight improvement with updates within BDSNP ('old' to 'new') and has similar dry conditions as central Texas.

We examined the performance of CMAQ under each of the soil NO parameterizations. Regions where soil NO parameterizations most impacted MDA8 ozone and $PM_{2.5}$ were examined for model performance in simulating CASTNET MDA8 $O_3$ and IMPROVE $PM_{2.5}$ observations.

For $PM_{2.5}$, our updated BDSNP module ('new') showed the best performance (Table 2). Evaluations against MDA8 $O_3$ observations found contrasting behavior for two different sets of CASTNET sites. The 11 mostly agricultural and prairie sites extending across the Midwest and southern US showed consistent overestimation as we moved from YL to BDNSP with new inputs, with bias jumping from ~ 7% to 14% and error from 15% to 19% (Table 3). However, the 5 forest/national park sites most of which lie near the San Joaquin Valley by contrast showed an overall improvement in bias from ~ 13% to 10% and in error from ~ 17 % to 15%  (Table 3).

Over-predictions of soil NO emissions especially in wet conditions may result from EPIC not properly accounting for on-farm nitrogen management practices like tile drainage. Crops such as alfalfa, hay, grass, and rice experience soil N loss due to tile drainage in wet soils (Gast et al., 1978; Randall et al., 1997). Recent updates to FEST-C (v. 1.2) include tile drainage for some crops but not hay, rice, grass and alfalfa (CMAS, 2015). Tile drainage results in loss of fertilizer N to water run-off from wet or moist soils.

We analyzed how the soil NO schemes affect the sensitivity of MDA8 ozone to anthropogenic $NO_x$ reductions by considering the 5.7 million tons/year reduction from 2011 levels that U.S. EPA expects for United States by 2025 with BAU scenario. These reductions were applied on basis of perturbation factors of relevant sectors keeping biogenic emissions unchanged for July 2011, based on EPA's annual baseline estimates between 2011 and 2025 (Table 4). These anthropogenic $NO_x$ reductions yield less reduction in MDA8 $O_3$ under the BDNSP soil NO scheme than YL, with 1-2 ppbv differences over parts of California and the Midwest (Figure 12). The shift occurs because our updated BDSNP schemes have higher soil NO in these regions, pushing them toward less strongly $NO_x$-limited regimes.



This work represents crucial advancement toward enhanced representation of soil NO in a regional model. Although possible wet biases and using dominant land cover rather than fractional in soil biome classification, may have over-predicted NO in agricultural regions in present study. The EPIC simulation used here lacks complete representation of farming management practices like tile, which can reduced soil moisture and soil NO fluxes. Inclusion of biogeochemistry influencing different reactive N species encompassing the entire N cycling could enable more mechanistic representation of emissions. For future work, there is a need for more accurate representation of actual farming practices and internalizing updated soil reactive N bio-geochemical schemes. More field observations are needed as well in order to increase the sample size for evaluation of modeled estimates soil emissions of reactive N species beyond NO.

## Code availability

The modified and new scripts used for implementation of BDSNP in CMAQ Version 5.0.2 are in the supplementary material. Also provided as supplement is the user manual giving details on implementing BDSNP module in-line with CMAQ, as used in this work. Source codes for CMAQ version 5.0.2 and FEST-C version 1.1 are both open-source, available with applicable free registration at http://www.cmascenter.org. Advanced Research WRF model (ARW) version 3.6.1 used in this study is also available as a free open-source resource at http://www2.mmm.ucar.edu/wrf/users/download/get_source.html.

## Acknowledgements

This work was supported by NASA's Air Quality Applied Sciences Team through a tiger team project grant for DYNAMO: DYnamic Inputs of Natural conditions for Air quality MOdels. Although this work was reviewed by EPA and approved for publication, it may not necessarily reflect official Agency policy.



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





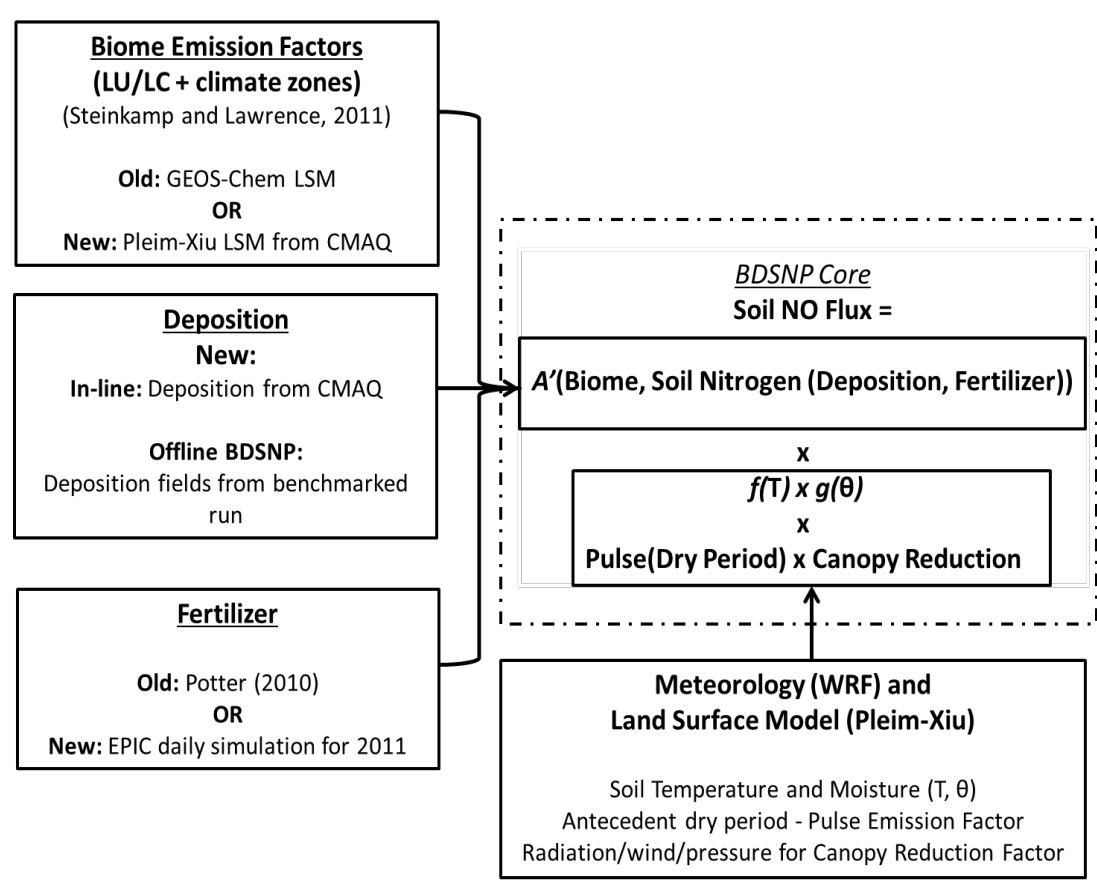



**Figure 1** Soil NO emissions modeling framework as implemented offline or in CMAQ (inline).
"Old" refers to the Hudman et al. (2012) implementation in GEOS-Chem. "New" refers to our
implementation in CMAQ.




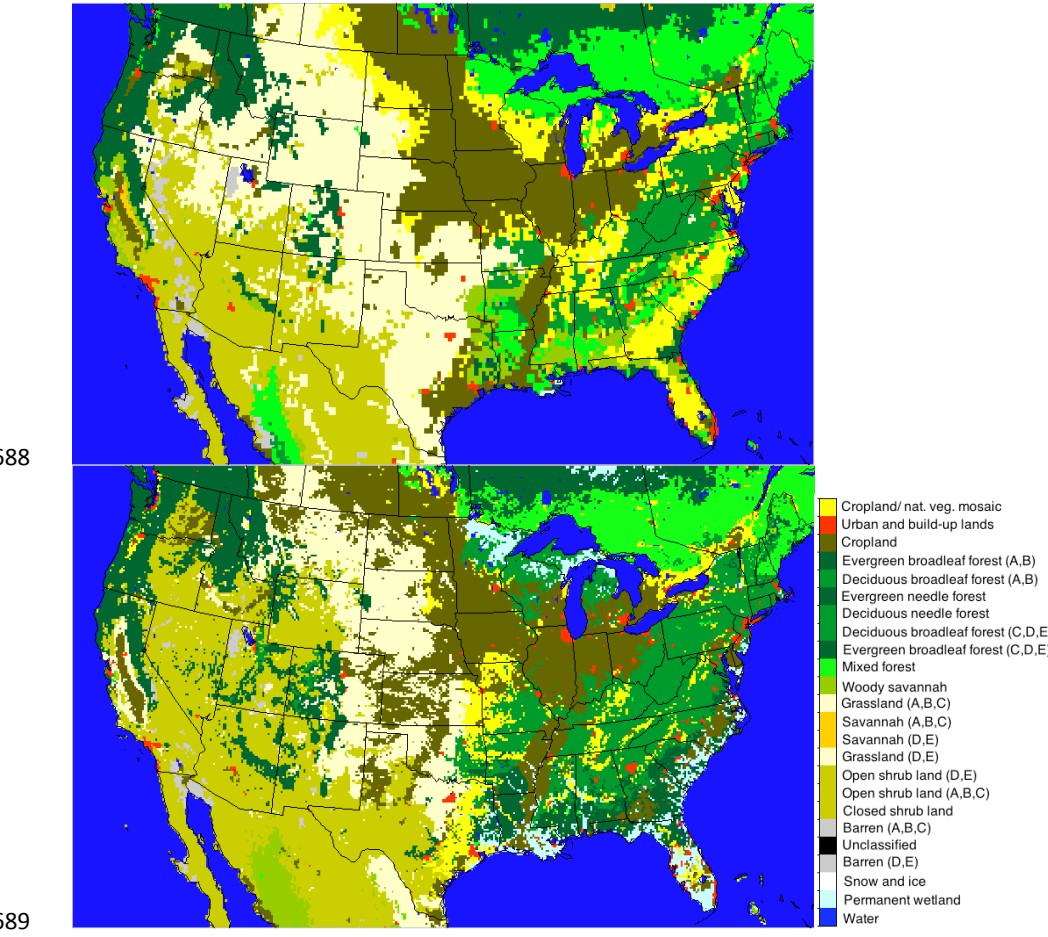



**Figure 2** Biomes from GEOS-Chem (0.25° x 0.25°; top) and CMAQ MODIS NLCD40 (12 km x 12 km; bottom) regrouped to match the classifications for which emission factors are available from Steinkamp and Lawrence (2011). See Tables A1 and A2 (right) for the mappings between classifications. The color-bar legends for classifications are as per NLCD definitions (http://www.mrlc.gov/nlcd11_leg.php).








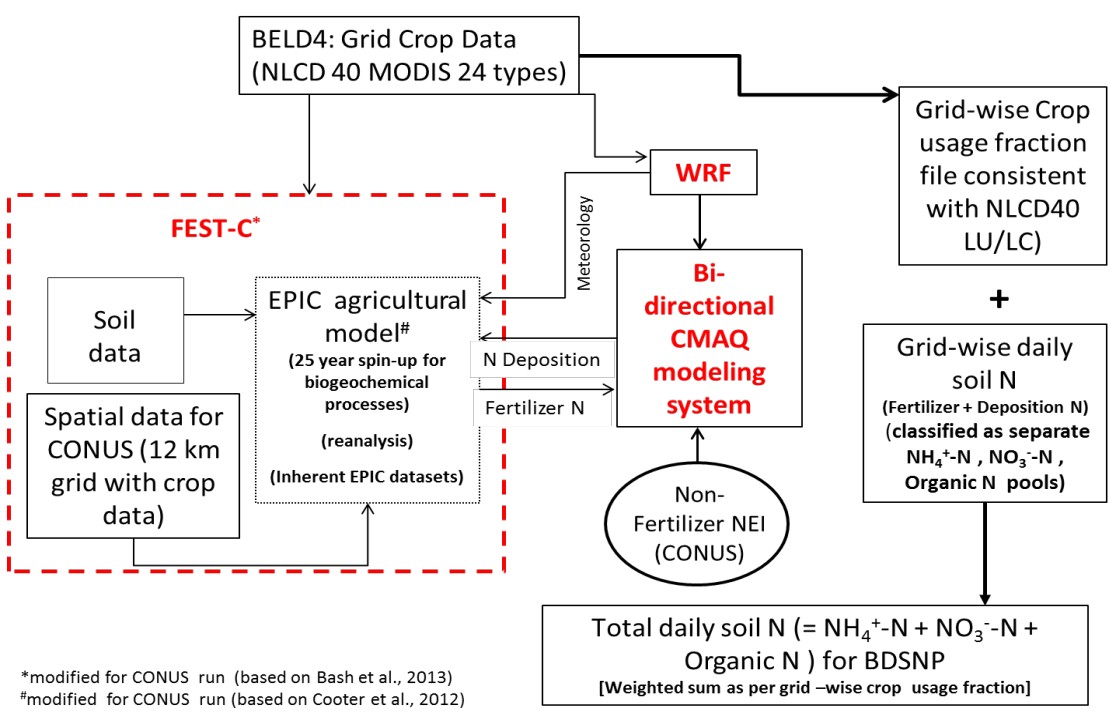


**Figure 3** Modeling framework for obtaining total soil N from EPIC using FEST-C.





Min (1, 1) = 0.0, Max (209, 164) = 76.8    Min (1, 1) = 0.0, Max (207, 211) = 115.5

**Figure 4** Potter (left) and EPIC (right) annual fertilizer application (Kg N/ha). Since EPIC
modeled only the U.S., Potter et al. (2010) is used in both cases to represent Canada and Mexico.



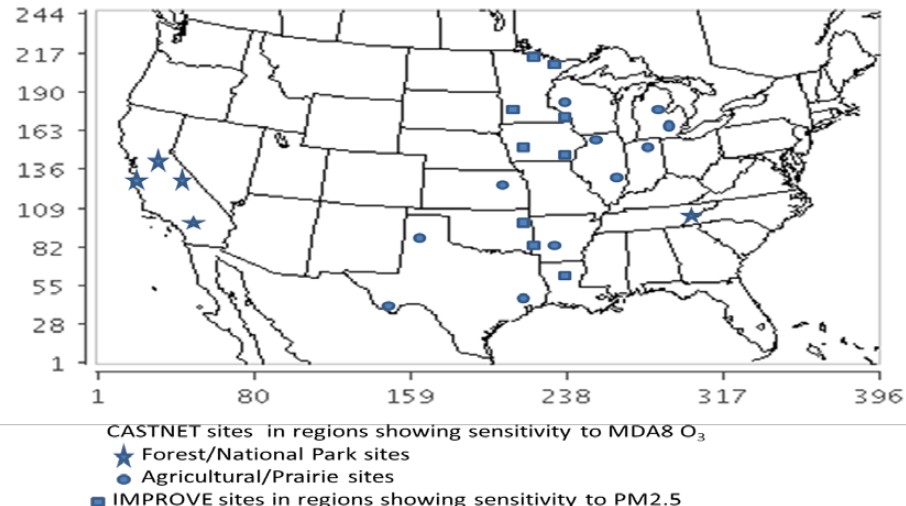


**Figure 5** CASTNET (Forest/National Park and agricultural sites) and IMPROVE sites in
continental US for comparison of modeled and observed ozone and PM$_{2.5}$.







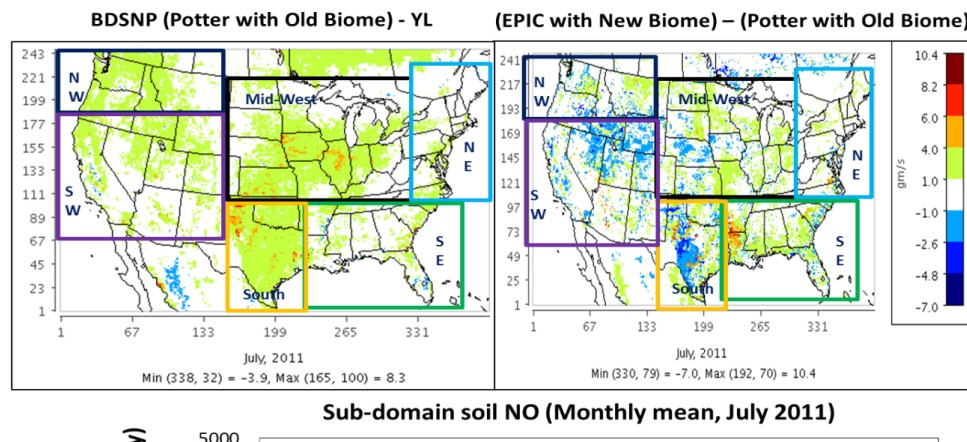

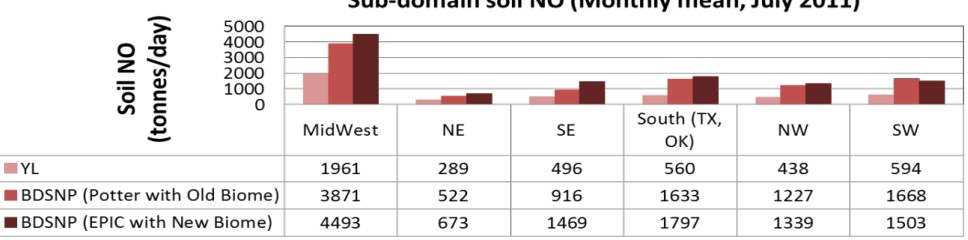


**Figure 6** Soil NO (tonnes/day) sensitivity to change from YL to BDSNP (Potter and old biome
or 'old') (left) and to the fertilizer and biome scheme within BDSNP (right) over sub-domains
(boxes).




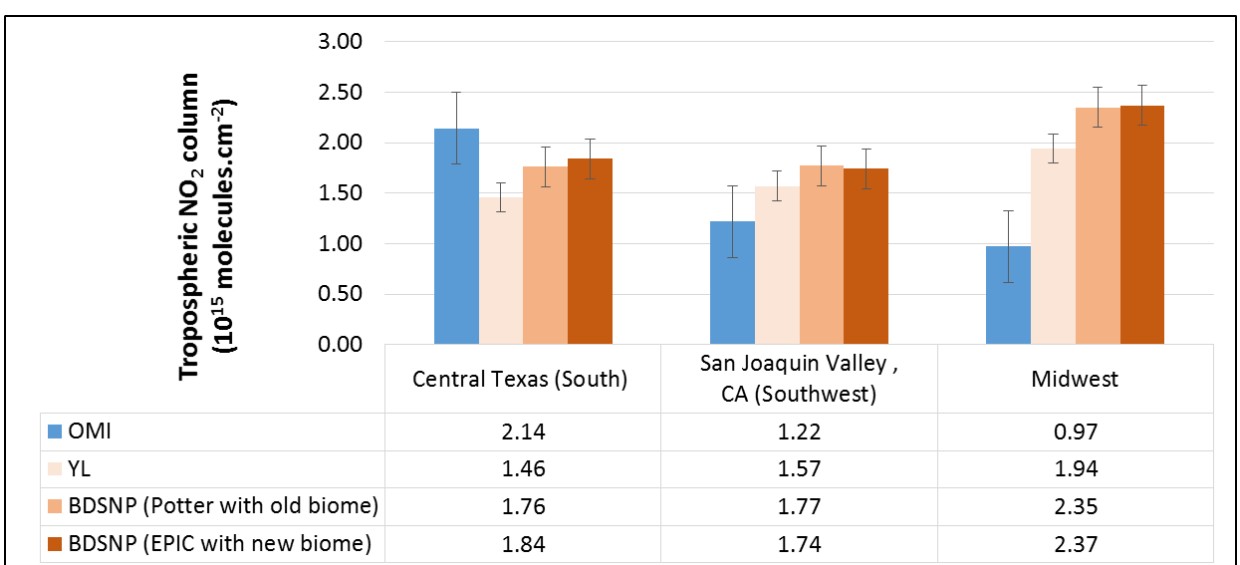


**Figure 7** Spatial average for Tropospheric $NO_2$ (molecules $cm^{-2}$) over regions with high soil NO sensitivity with switch from YL to BDSNP (as in Figure 6) with comparison to OMI $NO_2$. $NO_2$ column are temporal average for July 2011 at OMI overpass time.





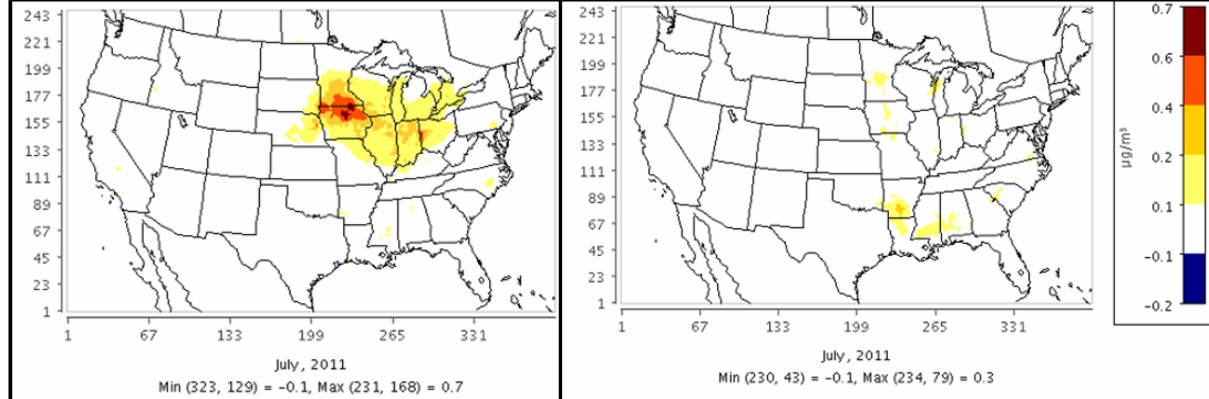


**Figure 8** Changes in modeled daily average PM$_{2.5}$ when switching from: a) YL to BDSNP
(Potter fertilizer data with original biome map) (left) and b) BDSNP (Potter with original
biomes) to BDSNP (EPIC with new biomes) (right).



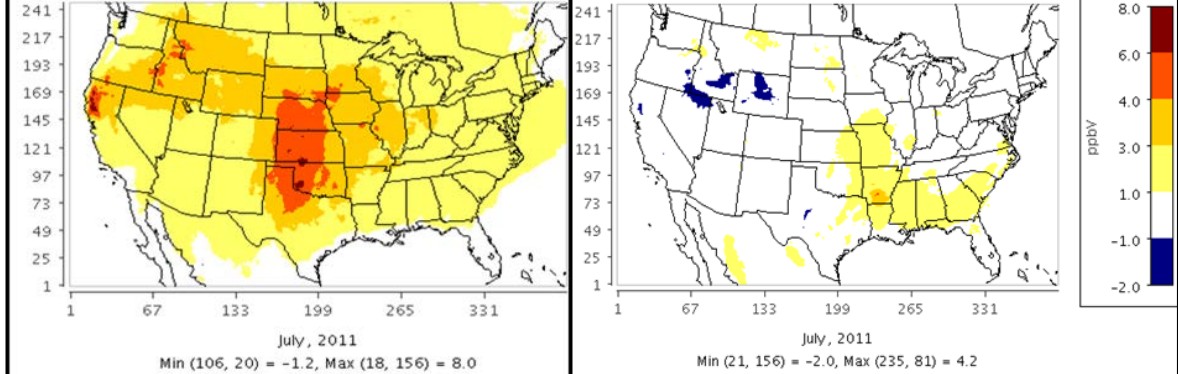


**Figure 9** Changes in modeled maximum daily 8-hour ozone (MDA8) when switching from: a)
YL to BDSNP (Potter fertilizer data with original biome map) (left) and b) BDSNP (Potter with
original biomes) to BDSNP (EPIC with new biomes) (right).






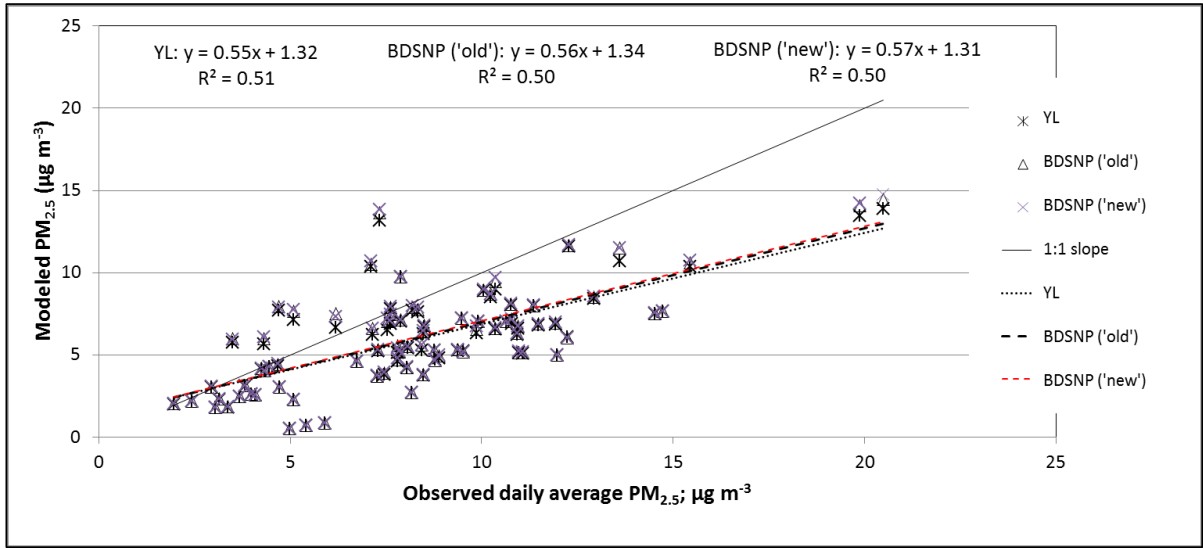


**Figure 10** Comparison of the three inline BDSNP-CMAQ cases with IMPROVE PM$_{2.5}$ data
(Malm et al., 1994) in continental US for Daily Average PM$_{2.5}$ for July 2011.









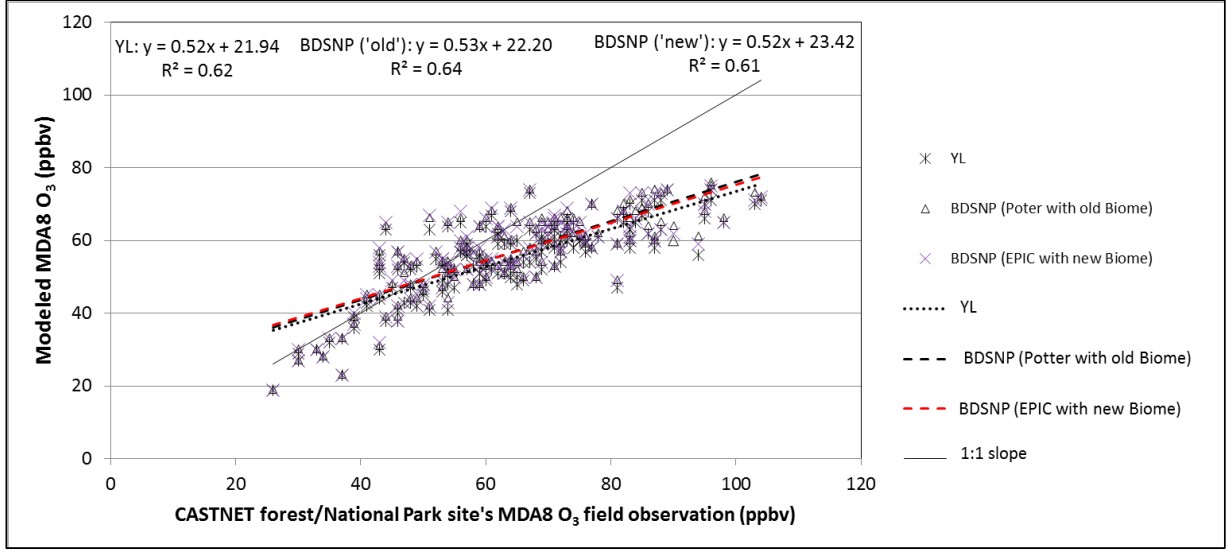


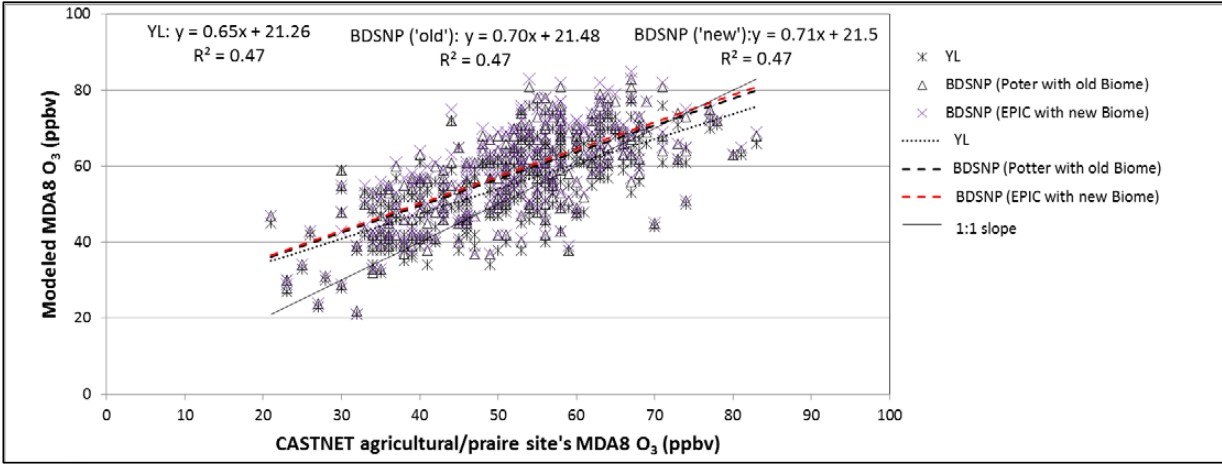


**Figure 11** Comparison of the three inline BDSNP-CMAQ cases with CASTNET MDA8 $O_3$ data
for forest/National Park sites in California (top, number of evaluation sites, n=147) and
agricultural/prairie sites in mid-west and south US (bottom, n=311) for July 2011.





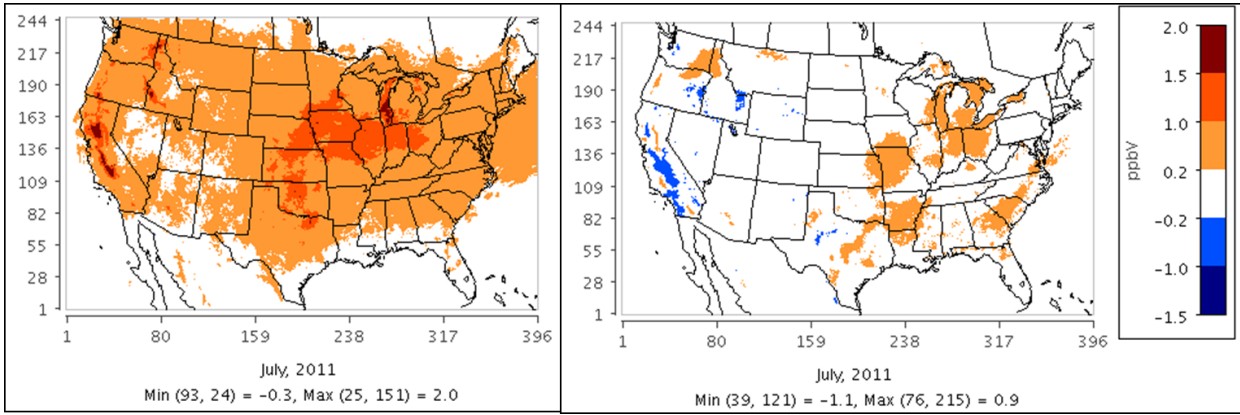


**Figure 12** Difference in monthly mean MDA8 $O_3$ perturbation between: a)    BDSNP ('old') –
YL (left) and, b) BDSNP ('new') – BDSNP ('old') (right). MDA8 $O_3$ perturbations are from
perturbed anthropogenic $NO_x$ estimates 2011 base case to 2025 base case, BAU (US EPA,
2015).

















**Table 1** Modeling configuration used for the WRF-BDSNP-CMAQ CONUS domain runs.

| WRF/MCIP | | | |
|---|---|---|---|
| **Version:** | ARW V3.6.1 | **Shortwave radiation:** | RRTMG scheme |
| **Horizontal resolution:** | CONUS (12kmX12km) | **Surface layer physic:** | Pleim-Xiu surface model |
| **Vertical resolution:** | 26 layer | **PBL scheme:** | ACM2 |
| **Boundary condition:** | NARR 32km | **Microphysics:** | Morrison double-moment scheme |
| **Initial condition:** | NCEP-ADP | **Cumulus parameterization:** | Kain-Fritsch scheme |
| **Longwave radiation:** | RRTMG scheme | **Assimilation:** | Analysis nudging above PBL for temperature, moisture and wind speed |
| **BDSNP** | | | |
| **Horizontal resolution:** | Same as WRF/MCIP | **Emission factor:** | Steinkamp and Lawrence (2011) |
| **Soil Biome type:** | 24 types based on NLCD40 (new) 24 types based on GEOS-Chem LSM (old) | **Fertilizer database:** | EPIC 2011 based from FEST-C (new) Potter et al. (2010) (old) |
| **CMAQ** | | | |
| **Version:** | V5.02 | **Anthropogenic emission:** | NEI2011 |
| **Horizontal resolution:** | Same as WRF/MCIP | **Biogenic emission:** | BEIS V3.1 in-line |
| **Initial condition:** | Pleim-Xiu (new) GEOS-Chem (old) | **Boundary condition:** | Pleim-Xiu (new) GEOS-Chem (old) |
| **Aerosol module:** | AE5 | **Gas-phase mechanism:** | CB-05 |

| Simulation Case Arrangement (in-line with CMAQ) | |
|---|---|
| 1. **YL:** | WRF/MCIP-CMAQ with standard YL soil NO scheme |
| 2. **BDSNP (Potter with old Biome or 'old'):** | WRF/MCIP-BDSNP-CMAQ with Potter and old biome |
| 3. **BDSNP (EPIC with new Biome or 'new'):** | WRF/MCIP-BDSNP-CMAQ with EPIC and new biome |

| Simulation Time Period | |
|---|---|
| | July 1-31, 2011 for CMAQ simulation with **in-line soil NO BDSNP module** |
| | Daily simulations in Year 2011 for **standalone BDSNP** soil NO BDSNP module (July 1-31, 2011 for sensitivity analysis) |

| Model Performance Evaluation |
|---|
| USEPA Clean Air Status and Trends Network (CASTNET) data for MDA8 ozone |
| Interagency Monitoring of Protected Visual Environments (IMPROVE ) Network (Malm et al., 1994) for $PM_{2.5}$ |
| OMI $NO_2$ satellite retrieval product as derived in Lamsal et al., 2014 for $NO_2$ column |





**Table 2** Aggregated performance statistics of CMAQ modeled daily average PM$_{2.5}$ for stations
showing sensitivities with change in soil NO between YL scheme and our 2 inline BDSNP
implementations ('old' and 'new') for CONUS in July 2011 as compared to observations at these
sites

| | Metrics | | | |
|---|---|---|---|---|
| | Sample Size | 81 | | |
| | Mean observed (µg/m$^3$) | 8.26 | | |
| | 3 CMAQ inline cases | YL | BDSNP (Potter with old biome) | BDSNP (EPIC with new biome) |
| Daily average PM$_{2.5}$ July (1 July- 31 July), 2011 | Mean predicted (µg/m$^3$) | 5.91 | 6.04 | 6.08 |
| | MAGE (Mean Absolute Gross error) | 2.86 | 2.80 | 2.77 |
| | RMSE | 3.45 | 3.40 | 3.38 |
| | R (correlation coefficient) Pearson's | 0.72 | 0.71 | 0.71 |
| | Spearman's Ranked | 0.65 | 0.63 | 0.63 |
| | NMB (%) | -28.52 | -26.90 | -26.44 |
| | NME (%) | 34.64 | 33.88 | 33.57 |






**Table 3** Performance statistics of CMAQ modeled MDA8 Ozone for 16 CASTNET remote sites
grouped into two categories: a) 11 sites with moist or wet soil condition (monthly mean soil
moisture (m$^3$/m$^3$), $\theta_{mean} > 0.175$), and b) 5 sites with dry soil condition ($\theta_{mean} < 0.175$) , using soil
NO from YL and our two inline BDSNP schemes.

| July 2011 | Metrics | | | |
|---|---|---|---|---|
| | Sample size | | 311 | |
| | Mean observed (ppbv) | | 51.76 | |
| | 3 CMAQ inline cases | YL | BDSNP (Potter with old biome) | BDSNP (EPIC with new biome) |
| 11 CASTNET sites (mostly agricultural/ prairie sites, Mostly wet soil conditions) | Mean modeled (ppbv) | 55.25 | 57.93 | 58.60 |
| | M$_{AGE}$ (Mean Absolute Gross error) | 7.78 | 9.16 | 9.65 |
| | RMSE | 9.41 | 10.96 | 11.47 |
| | R (correlation coefficient) — Pearson's | 0.50 | 0.51 | 0.50 |
| | Spearman's Ranked | 0.46 | 0.49 | 0.48 |
| | NMB (%) | 7.57 | 12.80 | 14.08 |
| | NME (%) | 15.65 | 18.38 | 19.33 |
| 5 CASTNET sites (mostly forest/National Park sites near San Joaquin valley CA, Dry soil conditions) | Sample size | | 147 | |
| | Mean observed (ppbv) | | 64.38 | |
| | Mean modeled (ppbv) | 55.17 | 57.01 | 56.87 |
| | M$_{AGE}$ (Mean Absolute Gross error) | 11.41 | 10.13 | 10.44 |
| | RMSE | 13.13 | 11.80 | 12.12 |
| | R (correlation coefficient) — Pearson's | 0.71 | 0.72 | 0.72 |
| | Spearman's Ranked | 0.68 | 0.69 | 0.69 |
| | NMB (%) | -13.14 | -10.23 | -10.35 |
| | NME (%) | 16.95 | 15.04 | 15.45 |




**Table 4** Emission perturbation factors applied to anthropogenic NO$_x$ emissions for each sector
listed in NEI as per EPA's RIA base-line reductions from 2011 to 2025 with BAU (Table 2A-1,
https://www3.epa.gov/ttn/ecas/docs/20151001ria.pdf)

| Sectors (NEI file names) | Perturbation factor |
|---|---|
| Electric Generating Unit(EGU)-point (ptimp- ptegu, ptegu_pk) | 0.7 |
| NonEGU-point (ptnonipm) | 1 |
| Point oil and gas (pt_oilgas) | 0.92 |
| Nonpoint oil and gas (np_oilgas) | 1.108 |
| Wild and Prescribed Fires (ptwildfire, ptprescfire) | 1 |
| Residential wood combustion (rwc) | 1.029 |
| Other nonpoint (nonpt) | 1.039 |
| Onroad (onroad) | 0.298 |
| Nonroad mobile equipment sources (nonroad) | 0.5 |
| Category 3 Commercial marine vessel (c3marine) | 0.77 |
| Locomotive and Category 1/Category 2 Commercial marine vessel (c1c2rail) | 0.62 |










# Appendix

**Table A1** List of 24 soil biome emission factor (EF) from Steinkamp and Lawrence (2010)

| ID | MODIS land cover | Köppen main climate[1] | EF1 (world geometric mean) | EF2 (world arithmetic mean) | EF3 (North American) |
|----|------------------|-----------------------|----------------------------|-----------------------------|----------------------|
| 1 | Water | -- | 0 | 0 | 0 |
| 2 | Permanent wetland | -- | 0 | 0 | 0 |
| 3 | Snow and ice | -- | 0 | 0 | 0 |
| 4 | Barren | D,E | 0 | 0 | 0 |
| 5 | Unclassified | -- | 0 | 0 | 0 |
| 6 | Barren | A,B,C | 0.06 | 0.06 | 0.06 |
| 7 | Closed shrub land | -- | 0.09 | 0.21 | 0.05 |
| 8 | Open shrub land | A,B,C | 0.09 | 0.21 | 0.09 |
| 9 | Open shrub land | D,E | 0.01 | 0.01 | 0.01 |
| 10 | Grassland | D,E | 0.84 | 1.05 | 0.62 |
| 11 | Savannah | D,E | 0.84 | 1.05 | 0.84 |
| 12 | Savannah | A,B,C | 0.24 | 0.97 | 0.24 |
| 13 | Grassland | A,B,C | 0.42 | 1.78 | 0.37 |
| 14 | Woody savannah | -- | 0.62 | 0.74 | 0.62 |
| 15 | Mixed forest | -- | 0.03 | 0.14 | 0.00 |
| 16 | Evergreen broadleaf forest | C,D,E | 0.36 | 0.95 | 0.36 |
| 17 | Deciduous broadleaf forest | C,D,E | 0.36 | 0.95 | 0.61 |
| 18 | Deciduous needle. forest | -- | 0.35 | 0.95 | 0.35 |
| 19 | Evergreen needle. forest | -- | 1.66 | 4.60 | 1.66 |
| 20 | Deciduous. broadl. forest | A,B | 0.08 | 0.13 | 0.08 |
| 21 | Evergreen broadl. forest | A,B | 0.44 | 1.14 | 0.44 |
| 22 | Cropland | -- | 0.57 | 3.13 | 0.33 |
| 23 | Urban and build-up lands | -- | 0.57 | 3.13 | 0.57 |
| 24 | Cropland/nat. veg. mosaic | -- | 0.57 | 3.14 | 0.57 |

(1). A-equatorial, B-arid, C-warm temperature, D-snow, E-polar (see Figure 2 for spatial map)



**Table A2** Mapping table to create the 'new' soil biome map based on NLCD40 MODIS land
cover categories

| ID | NLCD40 MODIS CATEGORY (40) | ID | SOIL BIOME CATEGORY (24) |
|---|---|---|---|
| 1 | Evergreen Needle leaf Forest | 19 | Evergreen Needle leaf Forest |
| 2 | Evergreen Broadleaf Forest | 16 and 21 | Evergreen Broadleaf Forest |
| 3 | Deciduous Needle leaf Forest | 18 | Dec. Needle leaf Forest |
| 4 | Deciduous Broadleaf Forest | 17 and 20 | Dec. Broadleaf Forest |
| 5 | Mixed Forests | 15 | Mixed Forest |
| 6 | Closed shrublands | 7 | Closed shrublands |
| 7 | Open shrublands | 8 and 9 | Open srublands |
| 8 | Woody Savannas | 14 | Woody savannah |
| 9 | Savannas | 11 and 12 | Savannah |
| 10 | Grasslands | 10 and 13 | Grassland |
| 11 | Permanent Wetlands | 2 | Permanent Wetland |
| 12 | Croplands | 22 | Cropland |
| 13 | Urban and Built Up | 23 | Urban and build-up lands |
| 14 | Cropland-Natural Vegetation Mosaic | 24 | Cropland/nat. veg. mosaic |
| 15 | Permanent Snow and Ice | 3 | Snow and ice |
| 16 | Barren or Sparsely Vegetated | 6 | Barren |
| 17 | IGBP Water | 1 | Water |
| 18 | Unclassified | 1 | Water |
| 19 | Fill value | 1 | Water |
| 20 | Open Water | 1 | Water |
| 21 | Perennial Ice-Snow | 3 | Snow and ice |
| 22 | Developed Open Space | 23 | Urban and build-up lands |
| 23 | Developed Low Intensity | 23 | Urban and build-up lands |
| 24 | Developed Medium Intensity | 23 | Urban and build-up lands |
| 25 | Developed High Intensity | 23 | Urban and build-up lands |
| 26 | Barren Land (Rock-Sand-Clay) | 24 | Cropland/nat. veg. mosaic |
| 27 | Unconsolidated Shore | 24 | Cropland/nat. veg. mosaic |
| 28 | Deciduous Forest | 16 and 21 | Evergreen Broadleaf Forest |
| 29 | Evergreen Forest | 19 | Evergreen Needle leaf Forest |
| 30 | Mixed Forest | 15 | Mixed Forest |
| 31 | Dwarf Scrub | 8 and 9 | Open shrublands |
| 32 | Shrub-Scrub | 8 and 9 | Open shrubland |
| 33 | Grassland-Herbaceous | 10 and 13 | Grassland |
| 34 | Sedge-Herbaceous | 14 | Woody savannah |
| 35 | Lichens | 10 and 13 | Grassland |
| 36 | Moss | 10 and 13 | Grassland |
| 37 | Pasture-Hay | 24 | Cropland/nat. veg. mosaic |
| 38 | Cultivated Crops | 22 | Cropland |
| 39 | Woody Wetlands | 2 | Permanent Wetland |
| 40 | Emergent Herbaceous Wetlands | 2 | Permanent Wetland |




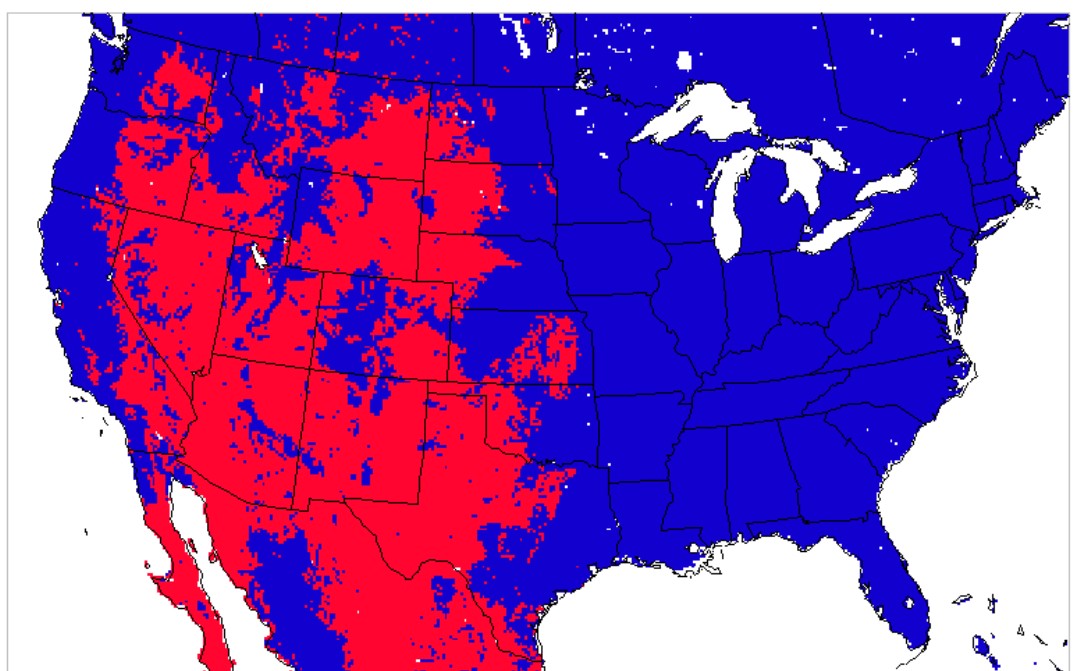


**Figure B1** Arid (red) and non-arid (blue) region over Continental US (12km resolution)


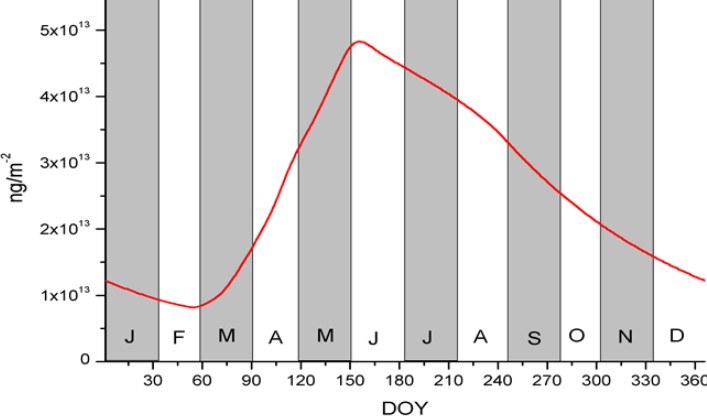


**Figure B2** Daily variation of total N from fertilizer application (from Potter et al. (2010)) processed from BDSNP to establish timing over continental US throughout 2011




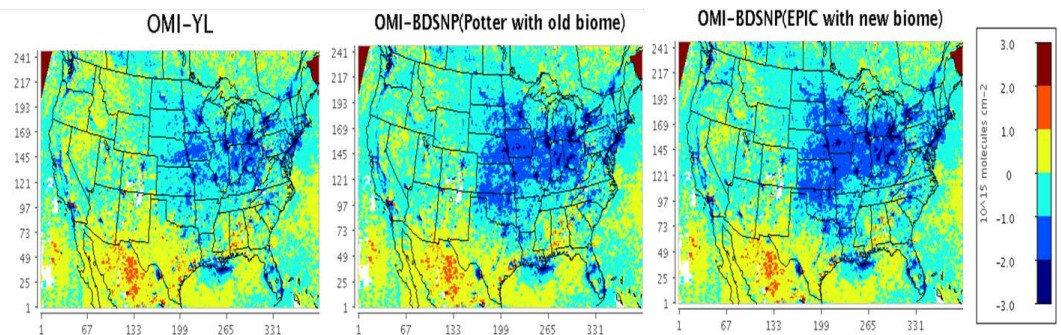


**Figure B3** Difference of OMI NO$_2$ column with NO$_2$ column simulated from the three inline CMAQ cases: YL, BDSNP (Potter with old biome), BDSNP (EPIC with new Biome) (left to right) over OMI overpass time averaged for July 2011 over CONUS. Note: In contour plots, white refers to gaps/no-fill values in OMI product and dark red at upper corners are due to gaps in CMAQ NO$_2$ column after temporal averaging at OMI overpass time.