# Peer review of "Enhanced representation of soil NO emissions in the"

_Geoscientific Model Development, 2016_

## Short Comment (SC1) · 6 Jun 2016

Dear authors,

In my role as Executive editor of GMD, I would like to bring to your attention our Editorial version 1.1:

http://www.geosci-model-dev.net/8/3487/2015/gmd-8-3487-2015.html

This highlights some requirements of papers published in GMD, which is also available on the GMD website in the 'Manuscript Types' section:

http://www.geoscientific-model-development.net/submission/manuscript_types.html

[Figure]

In particular, please note that for your paper, the following requirement has not been met in the Discussions paper:

- "The main paper must give the model name and version number (or other unique identifier) in the title."

Please add a version number for your model in the title upon your revised submission to GMD.

Yours,

Astrid Kerkweg

---

## Referee Comment (RC1) · Anonymous Referee #1 · 8 Jul 2016

This is a well done study that advances our understanding of regional modeling representation of soil NO emissions through the evaluation of improved (where appropriate: spatial and temporal) representation of soil conditions, fertilizer application, land use/land type, deposition, and meteorological influence. As regulation at the state and federal levels reduce the traditionally largest sources of NOx emissions (ie: power plants, mobile sources), other sources, like soil, will become relatively more important. Therefore, improvement of our understanding of the processes behind soil NO, and the relative contribution to air quality issues, is becoming increasingly important. I recommend this paper is published after a few minor edits.

Introduction: Lines 68-69: Suggest rewording this sentence for clarification. It seems to

suggest that deposition is a larger source of N in agricultural soils than fertilizer, which does not seem to be supported by the referenced papers.

The last two paragraphs of the introduction talk about your approach without specifically mentioning that you will be applying your updates to the CMAQ model (ie: you don't say you are running CMAQ). You should add that. Also, the text does not say whether you are running your simulation with bi-di (Figure 3 would suggest you are.) Can you clarify in the text?

Methodology: Lines 156-157: Awkward use of the word "significant". Are you trying to say that dry spring fertilizer application happens a lot?

Lines 164-165: make units consistent.

Lines 174-175: Personally curious, is there an existing or theoretical pathway through which this information could be used to reduce fertilizer demand and actual application? Would it be significant?

Lines 285-287: Do you say anywhere what the baseline year is for EPIC? (ie: land use, and management practices must be based on some start point? Or updated annually?)

Results and Discussion: Lines 356-357: this is in stark contrast to your introduction and suggested purpose for evaluating improved representation of key factors (ie: that NO emissions from soil are 1.5 to 4.5 times too low in the traditional YL representation). You discuss some reasons why (lines 436-438). Why would these factors not apply to other areas that are not over-estimated?

Lines 384-396: was met model performance for rain evaluated for this region in July? Ie: did you test this hypothesis? Since this also could help explain the over-estimation in the mid-west, it seems like this would be important to test.

---

## Referee Comment (RC2) · 11 Jul 2016

Better representation of soil NOx emissions in air quality models is important to have robust results of air quality modeling. This study by Rasool et al. is a good effort which builds upon a recently introduced parameterization to improve the timing and spatial distribution of soil NO emission estimates in the CMAQ model. I suggest the following revisions before this manuscript can be accepted for publication.

Comment 1. In Section 1 Introduction and Section 2 Methodology, the authors spend a lot of effort to describe the YL, BDSNP (Potter with old biome) and BDSNP (EPIC with new biome) schemes. To help readers better understand the differences between the three schemes, I would suggest the authors adding a table to summarize them.

Comment 2. Tables 2 & 3. For modeled daily average PM2.5 and MDA8 Ozone concentrations, the differences between the three schemes are very small. I would suggest the authors conducting a t-test to examine if the differences in the modeling results over the studying domain are statistically significant or not.

Comment 3. Pages 30-31. The quality of Figures 4 & 5 needs to be improved. The figures are stretched horizontally or vertically.

Comment 4. For Figure 10, the unit formats of x- and y- axis should be consistent. Currently, "PM2.5; $\mu$g m-3" is used for x-axis, but "PM2.5 ($\mu$g m-3)" is used for the y-axis.
* * *

---

## Author Comment (AC1) · 24 Aug 2016

We appreciate the generally favorable nature of the peer reviews and the opportunity to enhance the paper by responding to specific comments. Comments are shown in italics, with responses provided below, in the supplement file.

Please also note the supplement to this comment:
http://www.geosci-model-dev-discuss.net/gmd-2016-123/gmd-2016-123-AC1-supplement.pdf
* * *

---

## Author Response (AR1)

We appreciate the generally favorable nature of the peer reviews and the opportunity to enhance the paper by responding to specific comments. Comments are shown in italics, with responses provided below.

**Response to SC1: 'Executive Editor Comment':**

To comply with GMD guidelines, we added the version number to the title:

"Enhanced representation of soil NO emissions in the Community Multi-scale Air Quality (CMAQ) model version 5.0.2"

**Response to RC1: Anonymous Referee #1:**

*Lines 68-69: Suggest rewording this sentence for clarification. It seems to suggest that deposition is a larger source of N in agricultural soils than fertilizer, which does not seem to be supported by the referenced papers.*

We agree that the sentence was unclear, and have simplified the paragraph to the following for clarity:

> Both wet and dry deposition act as sources of nitrogen to soils (Yienger and Levy, 1995; Hudman et al., 2012). N is deposited in both oxidized (e.g., nitrate) and reduced (e.g., ammonium) forms, with ammonium representing a growing share of N deposition in the U.S. as anthropogenic $NO_x$ emissions are controlled (Li et al., 2016).

> Li, Y., Schichtel, B.A., Walker, J.T., Schwede, D.B., Chen, X., Lehmann, C.M., Puchalski, M.A., Gay, D.A. and Collett, J.L.: Increasing importance of deposition of reduced nitrogen in the United States. Proceedings of the National Academy of Sciences, p.201525736, 2016.

*The last two paragraphs of the introduction talk about your approach without specifically mentioning that you will be applying your updates to the CMAQ model (ie: you don't say you are running CMAQ). You should add that. Also, the text does not say whether you are running your simulation with bi-di (Figure 3 would suggest you are.) Can you clarify in the text?*

We clarify that the update is being applied to CMAQ (line 111) and that the bi-directional capability of CMAQ (which currently affects ammonia only) was applied (line 315-316).

*Methodology: Lines 156-157: Awkward use of the word "significant". Are you trying to say that dry spring fertilizer application happens a lot?*

We rephrased this to "is common practice".

*Lines 164-165: make units consistent.*

The units for mass have been made consistent.

*Lines 174-175: Personally curious, is there an existing or theoretical pathway through which this information could be used to reduce fertilizer demand and actual application? Would it be significant?*

Yes, N deposition in theory reduces the need for fertilizer and it is accounted for implicitly in EPIC's simulated farming practices.

*Lines 285-287: Do you say anywhere what the baseline year is for EPIC? (ie: land use, and management practices must be based on some start point? Or updated annually?)*

We clarify in Section 2.1.3 that 2011 is the baseline year and that EPIC updates land use and farm management practices annually based on the USDA Agricultural Resource Management Survey (ARMS) (Lines 174-177).

*Results and Discussion: Lines 356-357: this is in stark contrast to your introduction and suggested purpose for evaluating improved representation of key factors (ie: that NO emissions from soil are 1.5 to 4.5 times too low in the traditional YL representation). You discuss some reasons why (lines 436-438). Why would these factors not apply to other areas that are not over-estimated?*

In their global comparison of BDSNP-based soil NO emissions with an OMI satellite-based inversion, Vinken et al. (2014) found the central US to be one of the few areas where the BDSNP inventory over-estimated the satellite-based results (see Figures 6-7). This contrasted with other regions (Sahel, Australia and Eastern Europe) where Vinken found the BDSNP soil NO inventory to under-estimate OMI-based results. The reasons for these regional differences are still not well understood. (Lines 369-371)

*Lines 384-396: was met model performance for rain evaluated for this region in July? Ie: did you test this hypothesis? Since this also could help explain the over-estimation in the mid-west, it seems like this would be important to test.*

The figure below (included in supplementary material as Figure S6) evaluates modeled meteorological performance for precipitation compared to National Atmospheric Deposition Program (NADP) precipitation values for July 2011 on a monthly mean basis. Given the large site-to-site variability in WRF performance for precipitation, we remove the claim that a wet bias caused an over-estimation of soil NO.

[Figure]

**Fig. Normalized mean bias (%) of WRF model relative to NADP observed monthly mean precipitation for July 2011**

**Response to RC1: Reviewer 2:**

*Comment 1. In Section 1 Introduction and Section 2 Methodology, the authors spend a lot of effort to describe the YL, BDSNP (Potter with old biome) and BDSNP (EPIC with new biome) schemes. To help readers better understand the differences between the three schemes, I would suggest the authors adding a table to summarize them.*

We adopt the reviewer's suggestion by adding Table A3.

*Comment 2. Tables 2 & 3. For modeled daily average PM2.5 and MDA8 Ozone concentrations, the differences between the three schemes are very small. I would suggest the authors conducting a t-test to examine if the differences in the modeling results over the studying domain are statistically significant or not.*

We performed ANOVA (Analysis of variance) and t-test for MDA8 Ozone and daily average PM2.5 modeled concentrations as suggested by the reviewer. For MDA8 ozone, the differences between YL and BDSNP are significant (F ratio> F critical; $p<<0.05$), but the differences between the two BDSNP runs were not significant ($p>>0.05$). (Lines 376-378)

For MDA8 Ozone: F ratio > $F_{critical}$ and P-value << 0.05 from ANOVA analysis so the difference between 3 schemes are statistically significant.

Anova: Single Factor
(for MDA8 Ozone)

SUMMARY

| Groups | Count | Sum | Average | Variance |
|---|---|---|---|---|
| YL | 458 | 25237 | 55.10262 | 120.9107 |
| BDSNP (Potter with old Biome) | 458 | 26336 | 57.50218 | 136.8501 |
| BDSNP (EPIC with new Biome) | 458 | 26520 | 57.90393 | 140.6297 |

ANOVA

| Source of Variation | SS | df | MS | F | P-value | F crit |
|---|---|---|---|---|---|---|
| Between Groups | 2101.707 | 2 | 1050.854 | **7.913244** | **0.000383** | **3.002288** |
| Within Groups | 182064.4 | 1371 | 132.7968 | | | |
| Total | 184166.2 | 1373 | | | | |

| | One-tailed t-test P-value | 2-tailed t-test P-value |
|---|---|---|
| YL vs BDSNP(Potter with old biome) | 0.000714399 | 0.001428798 |
| YL vs BDSNP (EPIC with new biome) | 0.000111204 | 0.000222407 |
| BDSNP (Potter with old biome) vs BDSNP (EPIC with new biome) | 0.302939969 | 0.605879938 |

For PM2.5 (daily mean), F-ratio < $F_{critical}$ and p-value > 0.05 in comparisons among the three cases, so the differences are not that statistically significant.

Anova: Single Factor

SUMMARY

| Groups | Count | Sum | Average | Variance |
|---|---|---|---|---|
| YL | 81 | 478.456 | 5.906864 | 7.94423 |
| BDSNP (Potter with old Biome) | 81 | 489.285 | 6.040556 | 8.556796 |
| BDSNP (EPIC with new Biome) | 81 | 492.356 | 6.078469 | 8.771841 |

ANOVA

| Source of Variation | SS | df | MS | F | P-value | F crit |
|---|---|---|---|---|---|---|
| Between Groups | 1.316495 | 2 | 0.658247 | **0.078137** | **0.924861** | **3.033439** |
| Within Groups | 2021.829 | 240 | 8.424289 | | | |
| | | | | | | |
| Total | 2023.146 | 242 | | | | |

| | One-tailed t-test P-value | 2-tailed t-test P-value |
|---|---|---|
| YL vs BDSNP(Potter with old biome) | **0.383729** | **0.767459** |
| YL vs BDSNP (EPIC with new biome) | **0.353058** | **0.706116** |
| BDSNP (Potter with old biome) vs BDSNP (EPIC with new biome) | **0.467386** | **0.934773** |

*Comment 3. Pages 30-31. The quality of Figures 4 & 5 needs to be improved. The figures are stretched horizontally or vertically.*

Revised manuscript provides figures with better quality without distortion.

*Comment 4. For Figure 10, the unit formats of x- and y- axis should be consistent. Currently, "PM2.5; µg m-3" is used for x-axis, but "PM2.5 (µg m-3)" is used for the y-axis.*

In the revised version, parentheses are used for both unit formats (x- and y- axis).